# Rethinking Serialization in Linear 3D Vision: Decoupling Anisotropic Geometry from Isotropic Semantics

YinYun Yan [1 2]   Liping Zhang [1 3]   Tingran Wang [4]   Jiaxin Deng [5]   Changshuo Wang [6]   Limin Jiang [1]   Shanwei Gao [1]   Xin Ning [1 3]

## Abstract

Current linear State-Space Models (SSMs) for 3D point clouds typically rely on 1D serialization schemes (e.g., Hilbert curves) for global modeling. In dense scenes, such imposed order can disrupt spatial continuity and induce what we call serialization bias. We propose **AnIsoNet**, a framework that decouples anisotropic geometry from isotropic semantics via two dedicated modules: **L**ocal **A**nisotropy **G**eometric **M**odeling (LAGM) and **G**lobal **I**sotropy **S**emantic **A**ggregation (GISA). LAGM uses ellipsoidal encoding to capture local directionality without relying on global order. GISA is configured according to dataset-level geometric density: dense-scene datasets use Identity Mode to avoid additional geometry-driven re-serialization, whereas sparse-object datasets use Morton serialization to provide a lightweight spatial prior. This avoids redundant multi-view scanning while preserving $O(N)$ complexity. On S3DIS, reducing artificial serialization in dense scenes yields 82.62% mIoU, surpassing PCM by 3.0%. On ScanObjectNN, Morton serialization for sparse objects reaches 94.21% OA (+1.6%). On ScanNetV2, our model achieves 78.52% mIoU, surpassing PTv3 (77.5%) without pre-training, with only 12.2M parameters and 37G FLOPs.

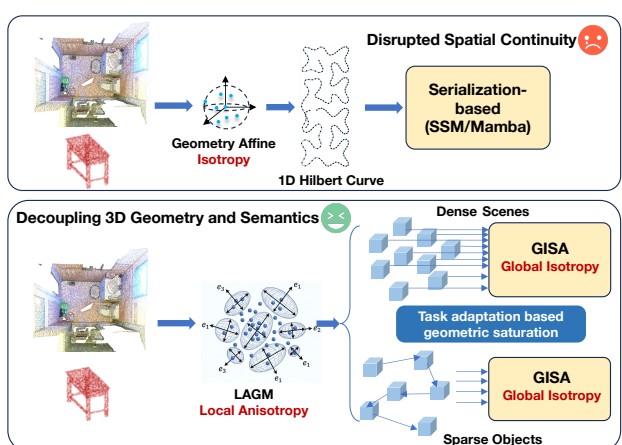

*Figure 1.* Architectural comparison of our method with serialization-based methods. Existing SSM/mamba methods force 3D point clouds into 1D sequences (top), introducing artificial ordering that disrupts spatial continuity. Our solution: decouple local anisotropic geometry from global isotropic semantics (bottom). LAGM captures directional features via ellipsoidal encoding, while GISA is configured according to data density—Identity Mode for dense scenes avoids explicit geometry-driven re-serialization, and Morton Mode provides spatial priors for sparse objects.

## 1. Introduction

Effective 3D point cloud understanding must reconcile local anisotropic geometry with global isotropic semantics, but the irregular and unordered nature of point sets makes this difficult. Early architectures, exemplified by PointNet++ (Qi et al., 2017b), prioritized local anisotropy through hierarchical grouping, yet struggled to maintain long-range global semantic coherence.

In contrast, Transformer-based methods such as Point Transformer (Zhao et al., 2021) and PTv2 (Wu et al., 2022) leverage the global receptive field of attention mechanisms, which better captures the semantic consistency associated with global isotropy. However, their $O(N^2)$ complexity severely limits efficiency. To address this, works such as Sonata (Wu et al., 2025) and PTv3 (Wu et al., 2024) introduced and refined Space-Filling Curves (SFCs) to serialize point clouds for lightweight processing. In that setting, the

---

[1] AnnLab, Institute of Semiconductors, Chinese Academy of Sciences, Beijing 100083, China [2] School of Electronic, Electrical, and Communication Engineering, University of Chinese Academy of Sciences, Beijing 101408, China [3] College of Materials Science and Opto-Electronic Technology, University of Chinese Academy of Sciences, Beijing 100049, China [4] China University of Mining Technology - Beijing, Beijing, China [5] Kuaishou Technology, Beijing, China [6] Department of Computer Science, University College London, London WC1E 6BT, UK. Correspondence to: Liping Zhang <zhangliping@semi.ac.cn>, Xin Ning <ningxin@semi.ac.cn>.

*Proceedings of the 43rd International Conference on Machine Learning*, Seoul, South Korea. PMLR 306, 2026. Copyright 2026 by the author(s).

1D order mainly serves as a memory layout rather than a causal processing path, allowing these models to retain point-to-point interactions across 3D space while improving efficiency.

Recent Linear State-Space Models (SSMs), represented by Mamba (Gu & Dao, 2024), offer a more attractive $O(N)$ solution. However, unlike Transformers that support non-causal attention, the strict recurrent path dependency of SSMs (where state $h_t$ strictly depends on $h_{t-1}$) introduces a new serialization bias. As illustrated in Figure 1, forcibly fitting 3D data into a 1D Hilbert or Morton sequence imposes an artificial causality that disrupts local neighborhoods and introduces directional noise, fundamentally violating the inherent isotropy of 3D environments. Current solutions typically resort to redundant "multi-view" ensembles (e.g., scanning point clouds 6+ times (Zhang et al., 2025)) to counteract this bias, thereby weakening the core efficiency advantage of SSMs.

For unordered 3D point clouds, geometry-driven serialization creates two coupled problems. First, it causes *topological rupture*: space-filling curves inevitably introduce jumps that break the local neighborhoods needed for anisotropic geometry modeling. Second, it creates *path dependency*: once the traversal becomes the recurrent processing path, sequence-local interactions introduce an artificial directionality into global semantic aggregation. We therefore argue that the role of serialization depends on geometric density. Dense scenes already provide strong local geometric context, whereas sparse objects can benefit from a lightweight spatial prior induced by serialization. Based on this observation, we propose AnIsoNet, which decouples local anisotropic geometry modeling from global semantic aggregation.

Our contributions are as follows:

- We identify serialization bias as a key bottleneck in 3D SSMs and propose a decoupling paradigm that addresses local geometry and global semantics with distinct topological assumptions.
- We design a dataset-aware GISA configuration that uses Identity Mode for dense scenes and Morton serialization for sparse objects.
- We achieve strong results on S3DIS (82.62% mIoU) and ScanObjectNN (94.21% OA), while on ScanNetV2 we reach 78.52% mIoU—surpassing PTv3 (77.5%) by 1.0% without external pre-training. Notably, our model uses only 12.2M parameters (26.4% of PTv3's 46.2M) and 37G FLOPs (vs. 180G).

**Conflict of Interest Disclosure.** The authors declare no financial conflicts of interest related to this work.

## 2. Related Work

### 2.1. Deep Learning on Point Clouds

Early architectures like PointNet++ (Qi et al., 2017b) established hierarchical sampling, while subsequent works explored diverse aggregation strategies (Wang et al., 2019). Recent point-cloud backbones further strengthen local geometric modeling and local-global decomposition through local geometric space coverage (Wang et al., 2022), acoustic-field fitting (Wang et al., 2025b), and global-perception/local-structure Transformer designs (Wang et al., 2024). For large-scale scenes, RandLA-Net (Hu et al., 2020) and sparse convolutions (Choy et al., 2019) introduced efficient processing pipelines. Despite their efficiency, these methods often lack global context modeling. Transformers bridged this gap, evolving from standard self-attention (Zhao et al., 2021) to Point Transformer V3 (PTv3) (Wu et al., 2024). PTv3 reduces the $O(N^2)$ complexity to $O(N \log N)$ by serializing points via Space-Filling Curves (SFCs). In PTv3, serialization mainly serves as locality-preserving hashing for patch construction, which is fundamentally different from the causal path dependency imposed by SSM-style sequence processing.

### 2.2. Linear Sequence Models & SSMs

State-Space Models (SSMs) have emerged as a powerful alternative to Transformers (Vaswani et al., 2017), offering linear $O(L)$ complexity. The foundational S4 model (Gu et al., 2022) introduced structured state spaces for long-range sequence modeling. Mamba (Gu & Dao, 2024) extends this with a selective state-space mechanism that makes $\mathbf{B}$ and $\mathbf{C}$ input-dependent while keeping the state transition matrix $\mathbf{A}$ as a learned parameter.

However, the fixed $\mathbf{A}$ still imposes exponential decay with sequence distance ($\mathbf{A}^{t-s}$), creating inherent bias toward sequential neighbors—acceptable for 1D sequences with natural ordering (text, audio) but problematic when the input lacks a natural sequential order. Concurrently, linear attention models like RWKV (Peng et al., 2023) and RetNet (Sun et al., 2024) reframe attention as recurrent processes. Gated Linear Attention (GLA) (Yang et al., 2024) further optimizes training efficiency via hardware-aware chunking. While effective for sequential data, these models are inherently causal and path-dependent, making their direct application beyond naturally ordered data non-trivial.

### 2.3. 3D SSMs and the Serialization Bottleneck

Inspired by Mamba, recent works have adapted SSMs for 3D vision. PointMamba (Liang et al., 2024) applies the selective state-space mechanism to point cloud classification and segmentation. PCM (Zhang et al., 2025) introduces multi-view serialization to reduce directional bias, achiev-

ing strong performance on S3DIS. Mamba3D (Han et al., 2024) proposes local feature enhancement via state-space modeling. StruMamba3D (Wang et al., 2025c) incorporates structural priors for improved shape understanding. PointR-WKV (He et al., 2025) adapts the RWKV architecture for hierarchical point cloud learning.

For 3D point clouds, which lack a natural sequential order, these approaches predominantly rely on the serialization strategy from PTv3 (e.g., Hilbert curves) to force the input into a 1D sequence compatible with SSMs' recurrent nature. This creates a tension we term serialization bias:

**Topological Rupture:** SFCs inevitably introduce jumps that sever local neighborhoods, disrupting the extraction of anisotropic geometric features (normals, curvature).

**Path Dependency:** The causal recurrence of SSMs ($h_t$ depends on $h_{t-1}$) imposes an artificial directionality that contradicts the isotropic nature of 3D space. Existing solutions often employ ensemble-based mitigation strategies; for instance, PCM employs multi-view serialization to reduce directional bias, albeit at the cost of increased computational inference ($6\times$). While effective, these approaches address the symptoms of serialization but do not fully decouple the underlying geometric-semantic conflict.

## 3. Method

### 3.1. Overview

Current 3D State-Space Models (SSMs) typically force point clouds into a single 1D sequence via space-filling curves, conflating local geometric modeling with global semantic aggregation. This serialization can introduce local neighborhood discontinuities that degrade performance in dense scenes. To address this limitation, we propose AnIsoNet, a unified framework that decouples these two processes (Figure 2). The framework consists of two complementary modules:

1. LAGM (Local Anisotropy Geometric Modeling): Encodes local micro-structure using direction-sensitive spectral encoding, independent of any global ordering.

2. GISA (Global Isotropy Semantic Aggregation): Aggregates global context through a unified decoder that supports two dataset-level modes: Identity Mode, which uses the default preprocessing/loading order without additional geometry-driven re-serialization, and Morton Mode, which uses Morton serialization for sparse-object inputs.

### 3.2. LAGM: Local Anisotropy Geometric Modeling

Our framework is built on a simple observation: 3D point cloud understanding requires local anisotropy for geometry

but global isotropy for semantics. At the local scale, geometric features are inherently directional—surfaces have orientations, edges have tangents, and curvature varies along specific directions. A planar surface exhibits strong variation along its normal direction but remains constant along the tangent plane. This directional dependency demands anisotropic encoding that respects the local coordinate frame defined by the surface geometry.

In contrast, global semantic aggregation should be isotropic: a "chair" remains semantically identical regardless of its position or orientation in the scene. This motivates a decoupled design in which LAGM handles local geometry and GISA handles global semantic aggregation, allowing the global module to focus on semantic evidence aggregation rather than additional geometric mixing.

The objective of LAGM is to capture local anisotropy within $k$-NN neighborhoods, independent of any global serialization order. For a point cloud $\mathcal{P} = \{p_i\}_{i=1}^N$ with coordinates $p_i \in \mathbb{R}^3$, the local neighborhood of point $p_i$ is defined as

$$\mathcal{N}_k(i) = \underset{\substack{\mathcal{S} \subseteq \{1,\ldots,N\} \setminus \{i\} \\ |\mathcal{S}| = k}}{\arg\min} \sum_{j \in \mathcal{S}} \|p_j - p_i\|_2^2. \qquad (1)$$

Here, $N$ is the number of points and $k$ is the neighborhood size. LAGM operates on this local graph and uses Ellipsoidal Spectral Encoding (ESE), where each channel acts as a learnable ellipsoidal template over the neighborhood.

**Ellipsoidal Spectral Encoding.** Standard relative position encodings typically treat offsets uniformly across directions. In contrast, ESE represents each channel $r$ as a learnable ellipsoidal prototype

$$\Theta_r = \{A_r, c_r\}, \qquad A_r = R_r \operatorname{diag}(s_r), \qquad (2)$$

where $r \in \{1, \ldots, C\}$ indexes the ESE prototype/channel, $C$ is the number of prototypes, $R_r$ controls the ellipsoid orientation, $s_r$ controls axis-wise scales, and $c_r \in \mathbb{R}^3$ is a learnable offset relative to the center point. Thus, $A_r \in \mathbb{R}^{3\times3}$ maps local offsets into the coordinate system of the $r$-th ellipsoidal template. For center point $p_i$ and neighbor $p_j \in \mathcal{N}_k(i)$, the geometric response is

$$d_{ij,r}^{\text{geo}} = \epsilon + \|A_r((p_j - p_i) - c_r)\|_2^2. \qquad (3)$$

Here, $\epsilon$ is a small constant for numerical stability, and $d_{ij,r}^{\text{geo}}$ is the squared ellipsoidal distance between neighbor offset $p_j - p_i$ and prototype $\Theta_r$. The neighborhood encoding is obtained by selecting the best-matching neighbor for each prototype:

$$n_{i,r} = \sqrt{\min_{j \in \mathcal{N}_k(i)} d_{ij,r}^{\text{geo}}}, \qquad n_i = [n_{i,1}, \ldots, n_{i,C}], \quad (4)$$

where $n_{i,r}$ is the response of prototype $r$ at point $p_i$ and $n_i \in \mathbb{R}^C$ is the local ESE descriptor. This formulation

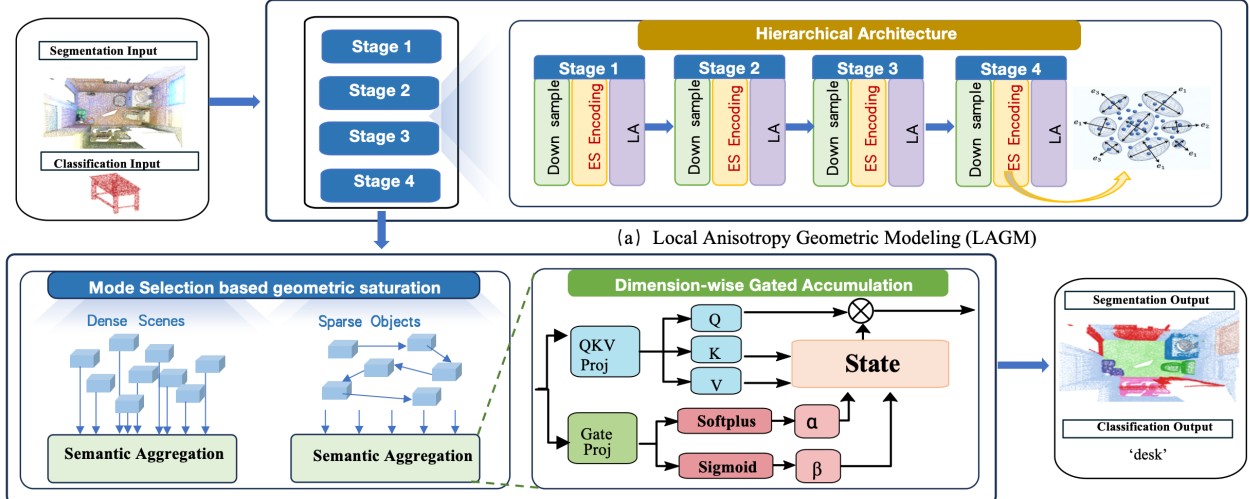

*Figure 2.* Overview of our AnIsoNet framework. (a) LAGM (Local Anisotropy Geometric Modeling) shows a representative hierarchical architecture; the number of stages is dataset-specific. Each stage combines sampling/downsampling, Ellipsoidal Spectral Encoding (ESE) to capture directional geometric features within k-NN neighborhoods, and Local Aggregation (LA) via geometry-gated max aggregation. (b) GISA (Global Isotropy Semantic Aggregation) aggregates global semantic information through dimension-wise gated accumulation. Left: dataset-level mode configuration based on geometric density—dense scenes (high point density) use the default preprocessing/loading order without additional geometry-driven re-serialization, while sparse objects (low density) use Morton serialization to inject a spatial prior. Right: gated state update mechanism with effective decay coefficient $\alpha$ and write gate $\beta$, enabling content-based retrieval through query projection. The framework supports both segmentation and classification tasks.

keeps the role of ESE simple: each channel measures how well the local neighborhood matches a learnable anisotropic ellipsoidal pattern. In the first stage, the same prototype-matching rule can be extended with lightweight input attributes, but the core mechanism remains geometric local matching.

**Local Aggregation.** The raw ESE response is projected to the feature space by

$$\tilde{n}_i = \mathrm{BN}(W_2\,\sigma(W_1 n_i))\,, \qquad (5)$$

where $W_1$ and $W_2$ are learnable linear projections, $\sigma(\cdot)$ denotes the activation function, and $\mathrm{BN}(\cdot)$ is batch normalization. The projected ESE feature is injected into the stage representation and then aggregated on the $k$-NN graph. The key point is that LAGM performs local anisotropic feature extraction before any global semantic aggregation.

**Hierarchical architecture.** Following the DeLA architecture (Chen et al., 2023), we use dataset-specific hierarchical LAGM encoders (Figure 2a). The figure shows a representative hierarchy, while the exact number of stages and channel widths depend on the benchmark.

### 3.3. GISA: Global Isotropy Semantic Aggregation

While LAGM captures local geometric features through anisotropic encoding, GISA focuses on global semantic

aggregation. Following the decoupling principle above, GISA avoids additional geometry-driven serialization in dense scenes and aggregates semantic evidence through a content-based linear mechanism. For sparse objects, it instead uses Morton serialization to inject a lightweight spatial prior.

**Mode Selection Based on Geometric Density.** In our current implementation, the GISA mode is configured at the dataset level rather than predicted per sample. We define a point cloud as *geometrically saturated* if local $k$-NN neighborhoods are sufficiently dense to reliably determine surface orientation. Formally, we measure local density via the average distance to $k$-nearest neighbors:

$$\sigma_k(p_i) = \frac{1}{k} \sum_{p_j \in \mathcal{N}_k(p_i)} \|p_j - p_i\|. \qquad (6)$$

A point cloud is geometrically saturated when $\sigma_k(p_i)$ is small across most points, indicating tightly packed neighborhoods. To enable meaningful comparison across datasets with different spatial scales, we normalize by the bounding box diagonal: $\hat{\sigma}_k = \sigma_k/\mathrm{diag}$. Empirically, dense scenes exhibit low normalized density (S3DIS: $\hat{\sigma}_k = 0.006$; ScanNetV2: $\hat{\sigma}_k = 0.004$), while sparse objects show significantly higher values (ScanObjectNN: $\hat{\sigma}_k = 0.025$, approximately $4\times$ larger). In our benchmarks, datasets with small $\hat{\sigma}_k$ are well served by Identity Mode because local

neighborhoods already capture substantial geometric context, whereas sparse objects benefit from a spatial prior introduced by serialization. We use this statistic as a dataset-level empirical guideline rather than a brittle per-sample classifier; Table 7 reports the measurements used in our experiments.

Based on this dataset-level criterion, we employ two complementary modes (see Figure 2b left):

- **Dense Scenes:** When local neighborhoods are already informative, we use the default preprocessing/loading order without additional geometry-driven re-serialization.

- **Sparse Objects:** When local neighborhoods are under-sampled, we use Morton serialization to inject a spatial prior.

Here, the Identity order should be understood as the default order after preprocessing and data loading, rather than as the scanner's raw acquisition order or a claimed canonical ordering.

**Dimension-wise Gated Accumulation.** Let $x_t \in \mathbb{R}^D$ denote the LAGM feature of the $t$-th point after the dataset-level ordering is chosen, where $D$ is the feature dimension and $t$ indexes the point sequence processed by GISA. Unlike standard DeltaNet (Yang et al., 2024) which maintains a matrix state $\mathbf{S} \in \mathbb{R}^{D \times D}$ to capture cross-dimensional interactions, we employ a *dimension-wise* vector state $\mathbf{s}_t \in \mathbb{R}^D$, initialized as $\mathbf{s}_0 = \mathbf{0}$, that accumulates information independently along each feature dimension:

$$\mathbf{s}_t = \mathbf{s}_{t-1} \odot \exp(-\alpha_t) + \beta_t \odot (\mathbf{k}_t \odot \mathbf{v}_t), \quad (7)$$

where $\odot$ denotes element-wise multiplication, $\alpha_t \in \mathbb{R}_+^D$ is the effective per-dimension decay coefficient induced by learned decay parameters and input-dependent activations, $\beta_t = \sigma(W_\beta x_t) \in \mathbb{R}^D$ gates the write strength, and $\mathbf{k}_t = \text{norm}(W_K x_t), \mathbf{v}_t = \text{norm}(W_V x_t) \in \mathbb{R}^D$ are L2-normalized key-value projections. The output is computed as:

$$\mathbf{o}_t = \mathbf{q}_t \odot \mathbf{s}_t, \quad (8)$$

where $\mathbf{q}_t = W_Q x_t \in \mathbb{R}^D$ retrieves information from the accumulated state via element-wise gating, and $W_\beta, W_K, W_V$, and $W_Q$ are learnable linear projections.

This form makes the retrieval mechanism explicitly content-driven: previous points contribute through the accumulated key-value evidence $\beta_t(\mathbf{k}_t \odot \mathbf{v}_t)$, and are selected by query-state matching rather than by a hand-designed serialization rule. In dense scenes, where local neighborhoods already capture substantial structure, this lets GISA aggregate broader semantic evidence without imposing an additional geometry-driven prior. For sparse objects, stronger

path-dependent decay under Morton serialization retains a lightweight spatial prior that remains helpful.

**Why Dimension-wise?** Since LAGM already models cross-point geometric interactions through $k$-NN aggregation, GISA only needs to aggregate long-range semantic evidence. We therefore use a vector state rather than a matrix state, reducing the state complexity from $O(D^2)$ to $O(D)$ while retaining content-based global accumulation.

**Why Mamba Needs Serialization but GISA Does Not.** A fundamental distinction lies in the architectural dependency on sequential ordering.

In Mamba (Gu & Dao, 2024), the state evolves as:

$$\mathbf{h}_t = \bar{\mathbf{A}} \cdot \mathbf{h}_{t-1} + \bar{\mathbf{B}}_t \cdot x_t, \quad y_t = \mathbf{C}_t \cdot \mathbf{h}_t, \quad (9)$$

where $\mathbf{h}_t$ is the recurrent hidden state, $\Delta_t$ is the input-dependent step size, $\mathbf{A}$ is the continuous-time transition parameter, $\bar{\mathbf{A}} = \exp(\Delta_t \mathbf{A})$ is the discretized transition matrix, and $\bar{\mathbf{B}}_t$ and $\mathbf{C}_t$ are input-dependent input/output maps. The recurrence $\mathbf{h}_t = f(\mathbf{h}_{t-1}, x_t)$ inherently requires a sequential ordering, so 3D point clouds must be artificially serialized and the contribution of $x_s$ to $y_t$ remains coupled to sequence distance. By contrast, GISA uses the content-based update in Eq. 7, where retrieval is driven by query-state matching rather than by sequence position induced by geometry-driven serialization. In dense scenes, the model tends to allocate more mass to high-retention states, whereas for sparse objects stronger path-dependent decay is preserved under Morton serialization. The small-decay limit therefore serves only as intuition for this high-retention regime; it should not be read as strict order invariance of intermediate states or outputs. This interpretation is consistent with the input-order perturbation results in Table 6; additional retention statistics are provided in Appendix Table 9.

**Parallel Implementation.** Although the GISA update has a recurrent form, it can be evaluated with a hardware-friendly chunk-wise scan over Eq. 7, preserving linear $O(N)$ complexity in sequence length. In our implementation, large point clouds are processed with a default maximum chunk size of 30K to control memory usage.

## 4. Experiments

### 4.1. Experimental Setup

**Datasets.** We evaluate AnIsoNet on three benchmarks spanning different geometric regimes: (1) S3DIS (Armeni et al., 2016) Area 5: Dense indoor scenes with approximately 100K points per room, suitable for evaluating isotropic aggregation. (2) ScanNetV2 (Dai et al., 2017): Large-scale indoor dataset with 1,513 annotated 3D scans.

*Table 1.* Semantic segmentation results on S3DIS Area 5. AnIsoNet achieves strong performance among linear-complexity methods.

| Method | Venue | Type | mIoU (%) |
|---|---|---|---|
| PointNet++ (Qi et al., 2017b) | NeurIPS'17 | MLP | 53.5 |
| Point Trans. (Zhao et al., 2021) | ICCV'21 | Trans. | 70.4 |
| PointNeXt-XL (Qian et al., 2022) | NeurIPS'22 | MLP | 74.9 |
| PTv2 (Wu et al., 2022) | NeurIPS'22 | Trans. | 72.7 |
| Stratified Trans. (Lai et al., 2022) | CVPR'22 | Trans. | 78.1 |
| Swin3D (Yang et al., 2025) | CVM'25 | Trans. | 74.5 |
| OctFormer (Wang, 2023) | SIGGRAPH'23 | Trans. | 76.6 |
| PointMamba (Liang et al., 2024) | AAAI'24 | SSM | 70.1 |
| PDNet (Yin et al., 2024) | AAAI'24 | MLP | 72.3 |
| OneFormer3D (Kolodiazhnyi et al., 2024) | CVPR'24 | Trans. | 72.4 |
| KPConvX (Thomas et al., 2024) | CVPR'24 | Conv | 73.5 |
| PTv3 (Wu et al., 2024) | CVPR'24 | Trans. | 73.4 |
| PTv3 + PPT | CVPR'24 | Trans. | 74.7 |
| DeepLA-120 (Zeng et al., 2025) | CVPR'25 | MLP | 75.7 |
| Sonata (Wu et al., 2025) (lin.) | CVPR'25 | Linear | 72.3 |
| Sonata (dec.) | CVPR'25 | Trans. | 74.5 |
| PCM (Zhang et al., 2025) | AAAI'25 | SSM | 79.6 |
| AnIsoNet (Ours) | - | Linear | 82.62 |

*Table 2.* Semantic segmentation results on ScanNetV2. Methods marked with † use external pre-training data. *Best among methods without pre-training. Underline denotes second-best without pre-training.

| Method | Venue | Type | Val (%) |
|---|---|---|---|
| PointNet++ (Qi et al., 2017b) | NeurIPS'17 | MLP | 53.5 |
| PointNeXt (Qian et al., 2022) | NeurIPS'22 | MLP | 71.5 |
| PTv2 (Wu et al., 2022) | NeurIPS'22 | Trans. | 75.4 |
| Stratified Trans. (Lai et al., 2022) | CVPR'22 | Trans. | 74.3 |
| Swin3D (Yang et al., 2025) | CVM'25 | Trans. | 76.4 |
| OctFormer (Wang, 2023) | SIGGRAPH'23 | Trans. | 77.1 |
| ConDaFormer (Duan et al., 2023) | NeurIPS'23 | Trans. | 75.1 |
| PointMamba (Liang et al., 2024) | AAAI'24 | SSM | 75.8 |
| HPENet (Zou et al., 2024) | AAAI'24 | MLP | 74.0 |
| OA-CNN (Peng et al., 2024) | CVPR'24 | Conv | 76.1 |
| OneFormer3D (Kolodiazhnyi et al., 2024) | CVPR'24 | Trans. | 76.6 |
| PTv3 (Wu et al., 2024) | CVPR'24 | Trans. | 77.5 |
| PTv3 + PPT† | CVPR'24 | Trans. | 78.6 |
| DeepLA-120 (Zeng et al., 2025) | CVPR'25 | MLP | 77.6 |
| Sonata (lin.) (Wu et al., 2025) | CVPR'25 | Linear | 72.5 |
| Sonata (dec.)† | CVPR'25 | Trans. | 79.1 |
| AnIsoNet (Ours) | - | Linear | 78.52* |

(3) ScanObjectNN (Uy et al., 2019) (PB_T50_RS): Sparse object point clouds with approximately 2K points per object, where spatial priors may benefit performance.

**Implementation.** Our encoder follows DeLA (Chen et al., 2023) with ellipsoidal spectral encoding. The GISA decoder uses a hidden dimension of $D = 512$ for scene-level tasks and $D = 384$ for object classification. Dense-scene datasets (S3DIS and ScanNetV2) use Identity Mode, whereas the sparse-object dataset ScanObjectNN uses Morton Mode. All experiments are conducted on a single NVIDIA RTX 3090 GPU.

### 4.2. Main Results

**S3DIS.** Table 1 reports results on S3DIS Area 5. AnIsoNet achieves 82.62% mIoU, the best result among the compared methods. Relative to linear-complexity baselines, it outperforms PCM (Zhang et al., 2025) by 3.0% and Sonata (lin.) by 10.3%. PCM mitigates serialization sensitivity through six-direction multi-view Mamba scanning, but still relies on repeated geometry-driven serializations. Relative to Transformer-based methods, our model exceeds PTv3 by 9.2%. This gap suggests that reducing geometry-driven serialization effects is especially beneficial in dense scenes.

This result is consistent with our claim that dense scenes benefit from avoiding additional geometry-driven serialization. Further qualitative analysis and cross-regime validation are provided in Section 4.3, while performance-efficiency comparisons are discussed in Section 4.4. Per-class results are given in Appendix Table 11.

**ScanNetV2.** On ScanNetV2 (Table 2), AnIsoNet achieves 78.52% mIoU on the validation set. Among linear-complexity methods, it outperforms Sonata (lin.) by 6.0%. Compared with PTv3, it is 1.0% higher on the validation split without external pre-training. Sonata (dec.) reaches 79.1%, which is 0.6% higher than our result, but it uses quadratic decoder attention together with large-scale external pre-training, whereas our model is trained entirely on the target dataset. This indicates that the dense-scene advantage of avoiding additional geometry-driven re-serialization extends beyond S3DIS to a larger and more diverse indoor benchmark. Per-class results are given in Appendix Table 12.

**ScanObjectNN.** Table 3 reports results on the challenging PB_T50_RS variant. AnIsoNet achieves 94.21% overall accuracy, the best result among the compared linear architectures without external pre-training. It outperforms Mamba3D (92.64%) by 1.6% and PointRWKV (93.63%) by 0.6%. In contrast to the dense-scene benchmarks, this sparse-object setting benefits from retaining a lightweight spatial prior through Morton serialization when local geometric context is limited. This complementary trend further supports the regime-dependent design of our decoupled framework. Per-class results are given in Appendix Table 13.

### 4.3. Analysis and Ablation

We next examine how the decoupled architecture and the regime-dependent serialization strategy contribute to the gains observed in the main benchmarks.

*Table 3.* Quantitative comparison on ScanObjectNN (PB_T50_RS). AnIsoNet outperforms recent MLP and SSM baselines.

| Method | Venue | Type | OA (%) | mAcc (%) |
|---|---|---|---|---|
| PointNet (Qi et al., 2017a) | CVPR'17 | MLP | 68.2 | 63.4 |
| PointCNN (Li et al., 2018) | NeurIPS'18 | Conv | 78.5 | 75.1 |
| DGCNN (Wang et al., 2019) | TOG'19 | Graph | 78.1 | 73.6 |
| GBNet (Qiu et al., 2022) | TMM'22 | Conv | 80.5 | 77.3 |
| PointMLP (Ma et al., 2022) | ICLR'22 | MLP | 85.4 | 83.9 |
| RepSurf-U (Ran et al., 2022) | CVPR'22 | Conv | 94.0 | - |
| PointNeXt-S (Qian et al., 2022) | NeurIPS'22 | MLP | 88.1 | 86.4 |
| PointVector (Deng et al., 2023) | CVPR'23 | Trans. | 88.2 | 86.7 |
| PointMeta (Lin et al., 2023) | CVPR'23 | Trans. | 88.1 | 86.9 |
| PointMamba (Liang et al., 2024) | AAAI'24 | SSM | 84.9 | 85.0 |
| Interpretable3D (Feng et al., 2024) | AAAI'24 | MLP | 88.0 | 86.5 |
| PDNet (Yin et al., 2024) | AAAI'24 | MLP | 88.5 | 86.8 |
| Mamba3D (Han et al., 2024) | ECCV'24 | SSM | 92.64 | - |
| StruMamba3D (Wang et al., 2025c) | ICCV'25 | SSM | 92.75 | - |
| PCM (Zhang et al., 2025) | AAAI'25 | SSM | 88.1 | 86.6 |
| DeepLA-24 (Zeng et al., 2025) | CVPR'25 | MLP | 90.6 | 89.5 |
| DyPolySeg (Wang et al., 2025a) | ICML'25 | Conv | 90.8 | - |
| PointRWKV (He et al., 2025) | AAAI'25 | RWKV | 93.63 | - |
| AnIsoNet (Ours) | - | Linear | 94.21 | 93.41 |

*Table 4.* Decoupling analysis on S3DIS. Δ is measured relative to the Sphere+MLP baseline.

| LAGM | GISA | mIoU (%) | Δ |
|---|---|---|---|
| Sphere | MLP | 73.48 | - |
| ✓ | MLP | 74.44 | +0.96 |
| Sphere | ✓ | 81.94 | +8.46 |
| ✓ | ✓ | 82.62 | +9.14 |

**Decoupling Analysis.** Table 4 shows that LAGM and GISA are complementary on S3DIS. Replacing the spherical local encoding with the ellipsoidal variant improves the baseline from 73.48% to 74.44% (+0.96%), while adding GISA alone yields a larger gain of 81.94% (+8.46%). Their full combination reaches 82.62%. This indicates that GISA provides the larger gain in this dense-scene setting, while LAGM still contributes a consistent improvement. The pattern is therefore complementary but asymmetric: in dense scenes, avoiding harmful global serialization effects is the dominant factor, while local anisotropic encoding remains a useful refinement. Appendix Table 10 shows the same complementary trend across all benchmarks.

**Serialization Strategy.** Figure 3 provides a qualitative explanation for why dense scenes prefer Identity Mode. For the same query point, Identity Mode keeps high-similarity responses concentrated on semantically coherent regions, whereas Morton Mode introduces elongated response bands aligned with the imposed traversal. Concretely, the Identity response remains localized around semantically matched regions, while the Morton response spreads along extended structures and forms stripe-like patterns that are less aligned with the true 3D neighborhood. This visualization suggests

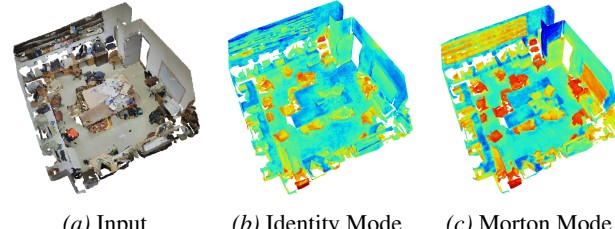

| *(a)* Input | *(b)* Identity Mode | *(c)* Morton Mode |
|---|---|---|

*Figure 3.* Feature-response visualization on S3DIS for the same query point. (a) Input. (b) Identity Mode produces more concentrated feature-similarity responses on semantically consistent regions. (c) Morton Mode can introduce stripe-like high-response patterns caused by geometry-driven serialization rather than true 3D neighborhood structure.

that geometry-driven serialization injects sequence-local correlations that are poorly matched to dense-scene semantics, consistent with the quantitative degradation observed when Morton serialization is applied to S3DIS.

**Cross-Regime Mode Selection.** Table 5 reports the mode-selection ablation across contrasting data regimes. On dense S3DIS, Identity Mode outperforms both Morton and Hilbert by about 8 points, showing that additional geometry-driven serialization is harmful when local neighborhoods are already sufficiently informative. On sparse ScanObjectNN, Morton Mode improves over Identity by 1.70 points, indicating that an explicit spatial prior becomes beneficial when local geometry alone is insufficient. A mismatched mode therefore causes noticeable degradation rather than collapse. Together, these results support configuring the global aggregation mode according to geometric density rather than enforcing a single serialization strategy across all point cloud settings.

*Table 5.* Cross-regime validation of dataset-level mode selection. Dense scenes prefer Identity Mode, while sparse objects benefit from spatial serialization.

| Dataset | Regime | Protocol | Identity/Default (%) | Hilbert (%) | Morton (%) |
|---|---|---|---|---|---|
| S3DIS | Dense scene | Mode ablation | 82.62 | 74.46 | 74.68 |
| ScanObjectNN | Sparse object | Mode ablation | 92.51 | 93.86 | 94.21 |

**Order Robustness in Dense Scenes.** Because our claim concerns robustness rather than strict permutation invariance, we directly test the task-relevant notion of robustness by perturbing the inference-time input order on ScanNetV2. Table 6 shows that predictions remain stable across reverse, lexicographic, Morton, Hilbert, random, and chunk-shuffled orders. Performance varies by at most 0.30 mIoU across substantially different input orders, indicating that dense-scene Identity Mode produces stable final predictions under order perturbations.

*Table 6.* Input-order perturbation on ScanNetV2.

| Input order | mIoU |
|---|---|
| Original | 78.47 |
| Reverse | 78.43 |
| Lexicographic | 78.69 |
| Morton | 78.39 |
| Hilbert | 78.65 |
| Random $\times$ 5 | $78.49 \pm 0.12$ |
| Chunk shuffle $\times$ 5 | $78.46 \pm 0.10$ |

**Density Criterion.** We also examine whether the dense/sparse distinction has a measurable geometric basis beyond qualitative interpretation. Table 7 reports the normalized neighborhood density statistic $\hat{\sigma}_k = \sigma_k/\text{diag}$ for the three benchmarks. In the current setting, the separation is clear: the two indoor scene datasets have substantially smaller normalized distances than ScanObjectNN, and their best-performing modes are correspondingly Identity for the indoor scenes and Morton for the sparse-object setting. We therefore treat $\hat{\sigma}_k$ as an empirical dataset-level heuristic for choosing an initial mode, rather than as a universal threshold.

*Table 7.* Geometric density statistics and preferred global aggregation mode.

| Dataset | $\hat{\sigma}_k$ | Best mode |
|---|---|---|
| S3DIS | 0.0060 | Identity |
| ScanNetV2 | 0.0042 | Identity |
| ScanObjectNN | 0.0251 | Morton |

## 4.4. Efficiency Analysis

We finally analyze efficiency from two complementary perspectives: parameter efficiency and actual computation cost.

Figure 4 and Table 8 provide two complementary views of efficiency on S3DIS. Figure 4 addresses parameter efficiency: AnIsoNet occupies a favorable region of the performance-parameter plane, achieving 82.62% mIoU with only 12.2M parameters. It lies well above larger SSM baselines such as PCM (79.6%, 38.5M) and substantially above larger Transformer baselines such as PTv3 (73.4%, 46.2M), indicating that the gain is not explained by parameter scaling alone. The comparison is also notable against similarly sized linear models: PointMamba uses 12.3M parameters but reaches only 70.1% mIoU. Overall, the figure shows that AnIsoNet improves accuracy while remaining in a much smaller parameter regime, rather than trading scale for performance.

Table 8 complements this with actual computation cost.

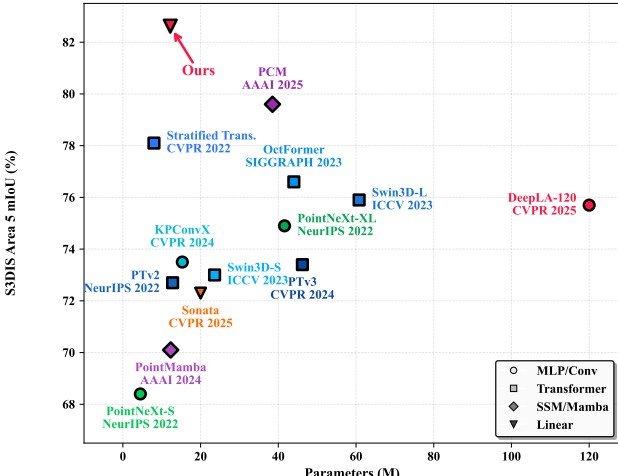

*Figure 4.* Parameter efficiency on S3DIS Area 5. AnIsoNet (red triangle) attains 82.62% mIoU with 12.2M parameters, outperforming larger models like PTv3 (46.2M, 73.4%) and PCM (38.5M, 79.6%). Shapes indicate method types (see legend). See Appendix Table 14 for detailed numerical values.

*Table 8.* Efficiency comparison on S3DIS Area 5.

| Method | Type | mIoU (%) | Params (M) | FLOPs (G) | Latency (ms) | GPU |
|---|---|---|---|---|---|---|
| PTv3 | Trans. | 73.4 | 46.2 | 180 | 142 | 8$\times$A100 |
| PCM | SSM | 79.6 | 38.5 | 95 | 89 | 4$\times$3090 |
| PointMamba | SSM | 70.1 | 12.3 | 42 | 68 | 1$\times$3090 |
| DeepLA-120 | MLP | 75.7 | 120.0 | 156 | 128 | 4$\times$A100 |
| Ours (Identity) | Linear | 82.62 | 12.2 | 37 | 98 | 1$\times$3090 |

On S3DIS, AnIsoNet uses 37G FLOPs, substantially lower than PTv3 (180G), PCM (95G), and DeepLA-120 (156G), while running on a single RTX 3090 GPU with a memory footprint of about 1.6G. Its latency is not the lowest in the table—PointMamba is faster per forward pass—but AnIsoNet offers a stronger accuracy-resource trade-off, combining clearly better accuracy with moderate runtime and without the multi-GPU requirements reported by several stronger baselines. This efficiency comes from GISA's chunk-wise scan over dimension-wise states, which avoids the heavier matrix-state computation used by DeltaNet-style linear attention while preserving expressive global aggregation.

## 5. Conclusion

This paper proposes AnIsoNet, a framework that rethinks serialization in 3D vision by decoupling local anisotropic geometry modeling from global isotropic semantic aggregation. Rather than using a uniform serialization strategy, local anisotropic features are first captured by LAGM, after which GISA is configured at the dataset level according to point cloud density—Identity Mode in dense scenes avoids addi-

tional geometry-driven re-serialization, while Morton Mode in sparse-object settings injects spatial priors. Through this decoupled design, our approach retains the efficiency of linear models while providing a more robust pathway for 3D representation learning.

**Limitations and Future Work.** This work has several limitations, which also point to directions for future research: 1) The Identity/Morton modes currently rely on dataset-level presetting, and mechanisms for dynamic switching based on input features remain to be explored; 2) The framework has been validated only on indoor scenes; its generalization capability on outdoor LiDAR point clouds with drastically different density distributions requires systematic evaluation; 3) Our training is fully supervised, and the synergy between large-scale self-supervised pretraining and the decoupled architecture has not been investigated—this could be key to unlocking further performance gains.

**Code Availability.** The implementation, training scripts, and benchmark configuration files are publicly available at github.com/yyy0218/anisonate.

## Acknowledgements

This work was supported by the National Natural Science Foundation of China under Grant Nos. 62373343 and 62403446. This work was also supported in part by the European Union's Horizon 2024 Research and Innovation Programme through the Marie Skłodowska-Curie Actions under Grant No. 101211118.

## Impact Statement

This paper presents work whose goal is to advance the field of machine learning. There are many potential societal consequences of our work, none of which we feel must be specifically highlighted here.

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

# A. Supplementary Results

This appendix reports additional measurements that support the main empirical findings, including GISA retention statistics, cross-module ablations, per-class results, and detailed efficiency comparisons.

## A.1. Additional Robustness and Ablation Results

### A.1.1. RETENTION BEHAVIOR OF GISA

Table 9 reports the empirical distribution of the per-step retention coefficient $r_t = \exp(-\alpha_t)$. Dense-scene Identity Mode assigns a larger fraction of states to the high-retention regime than Morton Mode, while ScanObjectNN remains dominated by low-retention states. These statistics complement the order-perturbation results in the main text and indicate that the effective decay behavior is dataset- and mode-dependent.

*Table 9.* GISA retention statistics. Fractions are computed over $r_t = \exp(-\alpha_t)$ bins.

| Setting | $r_t$ **mean** | $r_t \leq 0.1$ (%) | $0.1 < r_t \leq 0.9$ (%) | $r_t > 0.9$ (%) |
|---|---|---|---|---|
| S3DIS / Identity | 0.4313 | 55.18 | 2.97 | 41.84 |
| S3DIS / Morton | 0.1358 | 86.00 | 0.77 | 13.23 |
| ScanNetV2 / Identity | 0.3006 | 67.91 | 3.73 | 28.36 |
| ScanNetV2 / Morton | 0.1677 | 82.39 | 1.72 | 15.89 |
| ScanObjectNN / Identity | 6.03e-06 | 100.00 | 0.00 | 0.00 |
| ScanObjectNN / Morton | 5.94e-05 | 99.998 | 0.002 | 0.00 |

### A.1.2. CROSS-MODULE ABLATIONS ACROSS BENCHMARKS

Table 10 reports module-wise ablations on S3DIS, ScanNetV2, and ScanObjectNN. The contribution pattern varies by dataset: GISA gives the larger standalone gain on S3DIS and ScanObjectNN, whereas the LAGM-only variant is stronger than the GISA-only variant on ScanNetV2. Combining LAGM and GISA gives the best result on all three benchmarks, supporting their complementary roles.

*Table 10.* Cross-module ablation across benchmarks. ESE denotes the LAGM local encoder. S3DIS and ScanNetV2 report mIoU, while ScanObjectNN reports OA and mAcc.

| Dataset | Local | Global | Mode | mIoU/OA | mAcc |
|---|---|---|---|---|---|
| S3DIS | Sphere | MLP | – | 73.48 | – |
| S3DIS | ESE | MLP | – | 74.44 | – |
| S3DIS | Sphere | GISA | Identity | 81.94 | – |
| S3DIS | ESE | GISA | Identity | 82.62 | – |
| ScanNetV2 | Sphere | MLP | – | 75.88 | – |
| ScanNetV2 | ESE | MLP | – | 77.25 | – |
| ScanNetV2 | Sphere | GISA | Identity | 76.87 | – |
| ScanNetV2 | ESE | GISA | Identity | 78.52 | – |
| ScanObjectNN | Sphere | MLP | – | 90.35 | 89.08 |
| ScanObjectNN | ESE | MLP | – | 91.53 | 90.64 |
| ScanObjectNN | Sphere | GISA | Morton | 93.06 | 92.81 |
| ScanObjectNN | ESE | GISA | Morton | 94.21 | 93.41 |

## A.2. Extended Experimental Results

This section provides per-class results and a detailed efficiency comparison.

### A.2.1. DETAILED PER-CLASS RESULTS

Tables 11, 12, and 13 report per-class results on S3DIS Area 5, ScanNetV2 validation, and ScanObjectNN. The results provide category-level details for the aggregate metrics reported in the main text. On ScanObjectNN, Morton Mode improves overall accuracy over Identity Mode, consistent with the observation that sparse object inputs benefit from an explicit spatial prior.

*Table 11.* S3DIS Area 5 per-class mIoU. Comparison of different configurations. Identity Mode with ellipsoidal encoding achieves the best results. Note: Beam class has no samples in the Area 5 test set, hence 0.0 IoU.

| Method | mIoU | Ceil | Floor | Wall | Beam[†] | Col | Wind | Door | Table | Chair | Sofa | Book | Board | Clut |
|---|---|---|---|---|---|---|---|---|---|---|---|---|---|---|
| PTv3 + PPT | 74.7 | 95.2 | 98.1 | 88.5 | 0.0 | 65.2 | 66.8 | 88.2 | 92.5 | 95.1 | 91.2 | 89.7 | 91.3 | 82.4 |
| PCM | 79.6 | 94.8 | 97.9 | 87.2 | 0.0 | 63.1 | 64.5 | 86.9 | 91.8 | 94.6 | 90.5 | 88.4 | 90.1 | 81.2 |
| Ours (Morton) | 74.68 | 94.9 | 98.3 | 86.7 | 0.0 | 54.1 | 63.0 | 78.9 | 82.9 | 92.6 | 85.5 | 81.0 | 86.3 | 66.7 |
| Ours (Identity, Isotropic) | 81.94 | 97.6 | 98.8 | 92.9 | 0.0 | 74.6 | 74.0 | 85.0 | 94.5 | 96.4 | 89.9 | 90.8 | 84.3 | 86.5 |
| Ours (Identity, Ellipsoidal) | 82.62 | 96.7 | 98.8 | 89.7 | 0.0 | 71.3 | 69.1 | 91.0 | 94.2 | 96.5 | 93.9 | 92.4 | 94.9 | 85.6 |

[†] Beam class has no ground truth samples in S3DIS Area 5 test set.

*Table 12.* ScanNetV2 validation set per-class mIoU. Our method demonstrates balanced performance across diverse indoor categories.

| Method | mIoU | Wall | Flr | Cab | Bed | Chair | Sofa | Tabl | Door | Wind | Bksh | Pic | Cntr | Desk | Curt | Frid | Showr | Toil | Sink | Bath | Other |
|---|---|---|---|---|---|---|---|---|---|---|---|---|---|---|---|---|---|---|---|---|---|
| AnIsoNet | 78.52 | 88.5 | 96.6 | 72.9 | 81.9 | 91.6 | 80.5 | 80.8 | 71.9 | 71.6 | 86.4 | 39.7 | 73.5 | 73.5 | 84.0 | 74.4 | 78.2 | 96.0 | 70.4 | 90.3 | 67.8 |

*Table 13.* ScanObjectNN per-class accuracy. Comparison between Morton Mode (Ours) and Identity Mode. Morton serialization provides spatial priors that benefit most categories.

| Method | OA | Bag | Bin | Box | Cab | Chair | Desk | Disp | Door | Shelf | Table | Bed | Pill | Sink | Sofa | Toilet |
|---|---|---|---|---|---|---|---|---|---|---|---|---|---|---|---|---|
| Ours (Identity) | 92.51 | 72.3 | 90.0 | 94.0 | 94.1 | 98.0 | 87.3 | 95.1 | 95.7 | 93.0 | 83.7 | 90.0 | 95.2 | 90.8 | 97.1 | 96.5 |
| Ours (Morton) | 94.21 | 78.3 | 92.0 | 95.5 | 94.6 | 99.2 | 90.7 | 96.6 | 96.2 | 98.3 | 84.4 | 89.1 | 94.3 | 95.0 | 98.1 | 98.8 |

### A.2.2. COMPREHENSIVE METHOD COMPARISON ON S3DIS AREA 5

Table 14 compares representative methods on S3DIS Area 5 by architectural paradigm and provides the numerical values used in Figure 4. AnIsoNet reaches 82.62% mIoU with 12.2M parameters.

The comparison shows different accuracy-parameter trade-offs across model families. DeepLA-120 reaches 75.7% mIoU with 120M parameters, while transformer-based methods such as PTv3 and Swin3D-L use substantially larger models than AnIsoNet. Among linear and SSM-style methods, PCM obtains 79.6% mIoU with 38.5M parameters, and Sonata (lin.) obtains 72.3% mIoU with 20.0M parameters. These results provide additional context for the efficiency discussion in the main text.

*Table 14.* Comprehensive comparison of methods on S3DIS Area 5. This table provides detailed numerical data corresponding to Figure 4. Methods are categorized by architecture type and sorted by parameter count within each category.

| Method | Venue | Year | Type | Params (M) | mIoU (%) |
|---|---|---|---|---|---|
| *MLP/Conv-based Methods* | | | | | |
| PointNeXt-S | NeurIPS | 2022 | MLP | 4.5 | 68.4 |
| KPConvX | CVPR | 2024 | Conv | 15.3 | 73.5 |
| PointNeXt-XL | NeurIPS | 2022 | MLP | 41.6 | 74.9 |
| DeepLA-120 | CVPR | 2025 | MLP | 120.0 | 75.7 |
| *Transformer-based Methods* | | | | | |
| Stratified Trans. | CVPR | 2022 | Trans. | 8.0 | 78.1 |
| PTv2 | NeurIPS | 2022 | Trans. | 12.8 | 72.7 |
| Swin3D-S | CVM | 2025 | Trans. | 23.6 | 73.0 |
| OctFormer | SIGGRAPH | 2023 | Trans. | 44.0 | 76.6 |
| PTv3 | CVPR | 2024 | Trans. | 46.2 | 73.4 |
| Swin3D-L | CVM | 2025 | Trans. | 60.8 | 74.5 |
| *SSM/Mamba-based Methods* | | | | | |
| PointMamba | AAAI | 2024 | SSM | 12.3 | 70.1 |
| PCM | AAAI | 2025 | SSM | 38.5 | 79.6 |
| *Linear Attention Methods* | | | | | |
| Sonata (lin.) | CVPR | 2025 | Linear | 20.0 | 72.3 |
| AnIsoNet (Ours) | - | 2026 | Linear | 12.2 | 82.62 |

