# OpenReview forum: "Rethinking Serialization in Linear 3D Vision: Decoupling Anisotropic Geometry from Isotropic Semantics"
_ICML.cc/2026/Conference — ICML 2026 regular_

### Official Review · Reviewer_zWBA · 2026-02-23

**Soundness:** 3
**Presentation:** 3
**Significance:** 3
**Originality:** 3
**Overall Recommendation:** 4
**Confidence:** 1

**Summary:**

AnIsoNet addresses "Serialization Bias" in 3D State-Space Models (SSMs/Mamba), where forcing 3D data into 1D sequences disrupts spatial continuity. The framework decouples anisotropic local geometry from isotropic global semantics using two modules: Local Anisotropy Geometric Modeling (LAGM), which uses ellipsoidal spectral encoding, and Global Isotropy Semantic Aggregation (GISA), which adaptively switches between a "bias-free" Identity Mode for dense scenes and a Morton Mode for sparse objects.

**Compliance With Llm Reviewing Policy:**

Affirmed.

**Final Justification:**

The revisions have addressed my previous points, so I keep my initial rating that I am evaluating this paper from a broader perspective. Please note that since this topic lies outside my core area of expertise, my recommendations should be weighted accordingly.

**Key Questions For Authors:**

Please check the weakness part

**Limitations:**

Please check the weakness part

**Strengths And Weaknesses:**

I am not an expert in 3D data compression. Here are some Strengths And Weaknesses from my view:
Strengths:
+Efficiency: Restores the O(N) complexity advantage of SSMs by eliminating the need for redundant "multi-view" scans.
+Lightweight: Achieves state-of-the-art results with only 12.2M parameters—about 26% of Point Transformer v3's size.
+Adaptive Processing: The "Identity Mode" allows the model to process dense indoor scenes without artificial spatial ordering, preserving natural neighborhoods.

Weaknesses:
My only concern is that the performance seems highly relies on correctly identifying the geometric density regime. Could you show some visual cases when the method wrongly identifying the geometric density?

---

> ### Author Rebuttal · Authors · 2026-03-30
>
> We sincerely thank the reviewer for the encouraging review and for highlighting the core strengths of our work, particularly its $O(N)$ efficiency, lightweight design, and adaptive processing paradigm. Regarding the question, “What happens if the model wrongly identifies the geometric density regime, and are there visual cases of this?”, we respond by first clarifying how the mode selection is used in our framework, and then explaining the quantitative and qualitative consequences of a suboptimal choice.
>
> **1. Clarification: mode selection is a dataset-level configuration, not a runtime prediction.**
>
> To address the concern that the model might “misidentify” density on the fly, we clarify that the choice between `Identity Mode` and `Morton Mode` is not dynamically predicted for each sample during inference. Rather, it is used as a dataset-level configuration motivated by the empirical density analysis in the paper. As reported in Appendix Table 14, S3DIS = 0.0060, ScanNetV2 = 0.0042, and ScanObjectNN = 0.0251. Here, a smaller value of $\hat{\sigma}_k = \sigma_k / \mathrm{diag}$ means a smaller normalized average local k-NN distance, i.e., a denser point distribution; a larger value indicates a sparser regime. Under this interpretation, S3DIS and ScanNetV2 fall into the dense-scene regime, whereas ScanObjectNN falls into the sparse object-level regime. In this sense, dense scene datasets such as S3DIS and ScanNetV2 are treated with `Identity Mode`, whereas sparse object-level point clouds such as ScanObjectNN are better matched with `Morton Mode`. The role of the density statistic is therefore to provide an interpretable dataset-level prior, not to define a fragile per-sample decision rule.
>
> **2. Quantitative and qualitative consequences of a “wrong” configuration.**
>
> Our ablations already show that the cost of a mismatched mode is noticeable but not catastrophic. On dense S3DIS, Table 5 shows that `Identity` (82.62) clearly outperforms `Morton` (74.68). At the same time, 74.68 is still far from a collapse. As shown in Table 1, it remains competitive with a broad range of established baselines, exceeding methods such as PTv3 (73.4), KPConvX (73.5), OneFormer3D (72.4), PDNet (72.3), PointMamba (70.1), and Sonata (lin.) (72.3), while remaining close to several recent stronger baselines such as Sonata (dec.) (74.5), LitePT (74.6), and PTv3 + PPT (74.7). This suggests that even with a suboptimal serialization mode, the model can still partially compensate for the induced serialization bias through GISA's cross-path aggregation, i.e., by integrating information beyond a single imposed traversal path, and thus maintain reasonable functionality. A similar trend appears on sparse ScanObjectNN: using `Identity Mode` still gives 92.51 OA, compared with 94.21 OA for the best `Morton` setting. Thus, the effect of a wrong configuration is better characterized as a degradation from the optimum, rather than a brittle all-or-nothing failure.
>
> For qualitative understanding, the most direct evidence currently available in the paper comes from Figure 3 on S3DIS, where the same query point is fixed and each scene point is colored by its feature similarity to the query. Under `Identity Mode`, the response is more concentrated and semantically consistent, whereas under `Morton Mode`, the imposed spatial ordering introduces broken stripe-like high-response patterns because points that are distant in 3D space can become artificially adjacent along the traversal. To further clarify this point, please see the additional anonymous S3DIS visualizations here: [picture](https://anonymous.4open.science/api/repo/LAGM-GISA-Net-321C/file/img1.jpg?v=167b43d5). In these examples, the four panels from left to right are `RGB`, `ground truth`, `Identity Mode prediction`, and `Morton Mode prediction`. Consistently, in these dense indoor scenes, `Identity Mode` more often preserves fine local structures and semantic boundaries, while `Morton Mode` more often exhibits local over-smoothing, boundary leakage, or merging of nearby thin structures. For the sparse benchmarks, the tasks are mainly object-level classification, so analogous spatial visualizations are less natural; however, the quantitative results above show the same general phenomenon.
>
> We will revise the text to make this point clearer, and explicitly emphasize that the density analysis is a dataset-level statistical prior, at the scene or object level, rather than a fragile per-sample decision rule. We will also add more visual examples of suboptimal mode usage under dense and sparse settings in the supplementary material.
>
> We hope this addresses the concern, and we thank the reviewer again for helping us improve the clarity of the paper.

---

> > ### Author Rebuttal · Reviewer_zWBA · 2026-04-05
> >
> > Thanks for the authors' feedback. My concerns have been fully addressed.

---

> > > ### Author Response · Authors · 2026-04-05
> > >
> > > We sincerely thank the reviewer for the encouraging follow-up. We are grateful that our clarifications have fully addressed the concerns.

---

### Official Review · Reviewer_MQBU · 2026-03-12

**Soundness:** 4
**Presentation:** 4
**Significance:** 3
**Originality:** 3
**Overall Recommendation:** 5
**Confidence:** 5

**Summary:**

The paper studies serialization bias in linear 3D models for point clouds. The main claim is that forcing unordered 3D points into a 1D causal sequence, such as Hilbert or Morton order, can hurt dense-scene understanding by breaking local neighborhoods and imposing artificial directionality. To address this, the paper proposes AnIsoNet, which decouples local anisotropic geometry from global isotropic semantics using two modules: LAGM for local directional geometric encoding and GISA for global semantic aggregation. GISA uses Identity Mode in dense scenes and Morton Mode in sparse object settings. The headline empirical results are 82.62 mIoU on S3DIS, 78.52 mIoU on ScanNetV2, and 94.21 OA on ScanObjectNN, with 12.2M parameters and 37G FLOPs in the scene setting. Architecturally, the paper is quite clean. LAGM is a 4-stage hierarchical encoder with ellipsoidal spectral encoding over k-NN neighborhoods, while GISA is a dimension-wise gated accumulation module designed to avoid the strict sequential dependency of standard SSMs. The paper also proposes a density criterion based on normalized local neighbor distance, arguing that when the normalized density is low enough in dense scenes, global serialization becomes unnecessary or harmful, whereas sparse object clouds benefit from Morton ordering as a structural prior.

**Compliance With Llm Reviewing Policy:**

Affirmed.

**Key Questions For Authors:**

This is pretty a good paper in 3D community. How to linearize 3D data is quite essential to the community, both theoratically and practically. I have some questions below.

1. Since the paper argues that Identity Mode is serialization-bias-free or near permutation-invariant in dense scenes, can you report an explicit experiment with random shuffling of point order at inference time? That would directly test a central claim.
2. Is LAGM really essential, or is most of the gain from GISA? The ablation suggests the global semantic aggregation module provides most of the benefit. Can you provide stronger evidence for when the anisotropic local encoding materially matters beyond a relatively small additive gain?

**Limitations:**

As in weakness.

**Strengths And Weaknesses:**

### Strength
1. The paper identifies a real tension in 3D SSMs: causal sequence models need an order, but point clouds are unordered and semantically isotropic. The “serialization bias” framing is easy to understand and is more compelling than many papers that only present a new module without clarifying the actual failure mode they target.
2. The split between local anisotropic geometry (LAGM) and global isotropic semantics (GISA) is conceptually neat. The figure and the method section make the division of labor understandable: LAGM handles directional local structure, while GISA aggregates global semantic information without relying on the same kind of geometric ordering used by causal SSM baselines.
3. Very strong S3DIS result with good parameter efficiency.
The strongest empirical evidence is on S3DIS. The paper reports 82.62 mIoU, outperforming PCM by 3.0 points and PTv3 by 9.2 points while using far fewer parameters than PTv3. The efficiency plot also supports a strong performance/parameter tradeoff for this benchmark.
4. The dense-vs-sparse story is practically useful. The paper does more than claim a single universal mode is best. It gives a simple density-based rationale for why dense indoor scenes and sparse object clouds should be treated differently, and the benchmark results do follow that trend. That conditional message is probably the most valuable takeaway of the paper.


### Weakness
1. The “serialization-bias-free” and “permutation-invariant” claims feel too strong. The theory section argues that in dense scenes the learned decay can approach zero, making the accumulation effectively permutation-invariant. But Identity Mode still processes points in their original loading order, not under an exact permutation-invariant operator by construction. I did not see a direct experiment showing robustness to random point permutations, which would be important if the paper wants to make such a strong claim, or a closed mathematical proof.
2. Most of the improvement seems to be from the global semantic module, not the anisotropic geometry module. This is one of the clearest weaknesses. The ablation shows that ellipsoidal local encoding gives only a small gain, while adding GISA gives the large jump. That does not invalidate the method, but it does mean the title and framing may overemphasize “anisotropic geometry” relative to where the empirical benefit is actually coming from.

---

> ### Author Rebuttal · Authors · 2026-03-30
>
> We thank the reviewer for the careful reading, the positive assessment, and the constructive suggestions. The two main issues are also clear: how strongly we should describe Identity Mode, and how the gains should be attributed between global order handling and local anisotropic geometry. We clarify both points below.
>
> **1. What Identity Mode claims, and direct input-order perturbation evidence.**
>
> We agree that our original wording around “serialization-bias-free” or “near permutation-invariant” could be read more strongly than intended. Identity Mode should not be read as permutation-invariant by construction. The intended claim is narrower: it avoids explicit geometry-driven re-serialization and yields strong order robustness at the final prediction level in dense scenes.
>
> To examine this claim directly, we performed inference-time input-order perturbation experiments on ScanNetV2 with fixed model weights, changing only the input point order. The results are shown below.
>
> Table 1. Direct input-order perturbation test on ScanNetV2.
>
> | Input order | Runs | mIoU | mAcc | Acc |
> |---|---:|---:|---:|---:|
> | original | 1 | 78.47 | 86.77 | 92.57 |
> | reverse | 1 | 78.43 | 86.68 | 92.61 |
> | lex | 1 | 78.69 | 86.92 | 92.64 |
> | morton | 1 | 78.39 | 86.66 | 92.56 |
> | hilbert | 1 | 78.65 | 86.93 | 92.65 |
> | random × 5 | 5 | 78.49 ± 0.12 | 86.77 ± 0.09 | 92.61 ± 0.04 |
> | chunk_shuffle × 5 | 5 | 78.46 ± 0.10 | 86.73 ± 0.06 | 92.60 ± 0.04 |
>
> Here, `original` denotes the default preprocessing/loading order; `reverse` reverses it; `lex` sorts by `(x,y,z)`; `morton/hilbert` apply those serializations only at inference; and `random × 5` / `chunk_shuffle × 5` report mean ± std over five runs. The performance remains highly stable across substantially different input orders. While this does not establish strict permutation invariance of every intermediate state, it does directly support the weaker, task-relevant claim that Identity Mode yields strong order robustness in dense scenes.
>
> Accordingly, we will revise the wording from “serialization-bias-free” / “permutation-invariant” to the more accurate claim that Identity Mode avoids geometry-driven serialization priors and yields strong order robustness in dense scenes. We will make this change in the Abstract, Figure 1, the Introduction contribution summary, and the GISA description in the Method section.
>
> **2. The respective roles of LAGM and GISA.**
>
> We also agree that this is an important question, and that the current main-paper presentation can give the impression that most of the gain comes from GISA. In that sense, the reviewer’s reading is reasonable. At the same time, the broader ablations suggest a more nuanced conclusion: GISA and LAGM address different sources of error and both contribute materially, with the relative gain depending on the benchmark and regime.
>
> Table 2. Cross-module ablation on ScanNetV2.
>
> | LAGM | GISA | Global Head / Mode | mIoU |
> |---|---|---|---:|
> | ✗ | ✗ | MLP | 75.88 |
> | ✓ | ✗ | MLP | 77.25 |
> | ✗ | ✓ | Identity | 76.87 |
> | ✓ | ✓ | Identity | 78.52 |
>
> Table 3. Cross-module ablation on ScanObjectNN.
>
> | LAGM | GISA | Mode | OA | mAcc |
> |---|---|---|---:|---:|
> | ✗ | ✗ | MLP | 90.35 | 89.08 |
> | ✓ | ✗ | MLP | 91.53 | 90.64 |
> | ✗ | ✓ | Morton | 93.06 | 92.81 |
> | ✓ | ✓ | Morton | 94.21 | 93.41 |
>
> These results show that the pattern is dataset-dependent rather than one-sided. On ScanNetV2, the standalone gain of LAGM is already substantial: the baseline improves from 75.88 to 77.25 (+1.37 mIoU), whereas GISA-only gives 76.87 (+0.99 mIoU). Thus, on this benchmark, local anisotropic geometry plays a clear role. On ScanObjectNN, the pattern is different and more aligned with the reviewer’s observation: the larger jump comes from GISA-only (93.06/92.81), while LAGM still improves the MLP baseline (90.35/89.08 -> 91.53/90.64) and further improves the GISA-based model to 94.21/93.41. This is also consistent with the S3DIS ablation in the paper, where replacing the spherical local design with ESE improves 73.48 -> 74.44.
>
> Therefore, the more accurate framing is not that the two modules contribute equally, but that they play complementary and asymmetric roles: GISA mainly improves global aggregation under serialization bias, while LAGM provides a stable, and in some cases substantial, gain from local anisotropic geometry across benchmarks. We will revise the contribution summary and the ablation discussion accordingly, so that this asymmetry is stated explicitly.

---

> > ### Author Rebuttal · Reviewer_MQBU · 2026-04-03
> >
> > Thank you for the detailed rebuttal and for running the additional experiments. The new evidence is helpful and resolves part of my concerns. In particular, the direct input-order perturbation test on ScanNetV2 is useful: performance remains very stable across original, reverse, lexicographic, Morton, Hilbert, random, and chunk-shuffled orders, which supports a weaker and more precise claim of strong order robustness in dense scenes. The added cross-module ablations are also informative. On ScanNetV2, LAGM-only improves the baseline from 75.88 to 77.25 mIoU, compared with 76.87 for GISA-only, while the full model reaches 78.52. On ScanObjectNN, GISA provides the larger jump, but LAGM still improves both the baseline and the GISA-based model, which makes the module attribution story clearer and more convincing.
> >
> > However, I still think the paper’s central framing should be toned down. The new results support a narrower conclusion than the original wording: Identity Mode is not permutation-invariant by construction, but rather robust to input order in dense scenes; similarly, LAGM and GISA play complementary but asymmetric roles, rather than supporting a broad claim that the serialization issue is generally resolved. In that sense, the rebuttal improves my confidence in the empirical findings, but it does not fully remove my concern that the main concept is somewhat overclaimed, especially around phrases such as “serialization-bias-free” and the broader generality of the contribution. Overall, the rebuttal is constructive and appreciated, but my overall assessment remains unchanged.

---

> > > ### Author Response · Authors · 2026-04-04
> > >
> > > We sincerely thank the reviewer for the recognition, positive assessment, and helpful follow-up suggestions, which have been very valuable in improving the quality of our paper. We fully agree with the reviewer's comments regarding the precision of our wording and the appropriate scope of our claims. These suggestions help us state our claims more accurately and rigorously, and we will revise the manuscript accordingly to better align the presentation with the actual scope of our evidence.
> > >
> > > More specifically, we agree that the additional experiments support a narrower claim than the original wording. What they support is that, in dense scenes, `Identity Mode` avoids additional geometry-driven re-serialization and shows strong order robustness at the final prediction level, rather than being permutation-invariant by construction.
> > >
> > > Concretely, we will revise `Identity Mode (serialization-bias-free)` in `Figure 1` to `Identity Mode (without additional geometry-driven re-serialization)`; revise `enforcing permutation invariance in global semantic aggregation` in the `Introduction` to `promoting order-robust global semantic aggregation in dense scenes`; revise `serialization-bias-free Identity Mode` in the contribution summary to `Identity Mode that avoids explicit geometry-driven re-serialization`; and similarly replace phrases such as `permutation-invariant aggregation` and `Serialization-Bias-Free Aggregation` in the `Method section` with more cautious wording such as `order-robust aggregation in dense scenes`, to better reflect the actual scope of our evidence. We will also revise the theoretical wording around the $\alpha \to 0$ discussion so that it is presented only as an intuition for why order sensitivity may be reduced in dense scenes, rather than as a proof of permutation invariance or a strict guarantee of order-independent aggregation.
> > >
> > > We will also revise the descriptions of LAGM and GISA to more accurately reflect their complementary yet asymmetric roles: LAGM provides local anisotropic geometric encoding and yields stable gains, particularly in dense-scene benchmarks, while GISA plays the larger role in global order handling, with especially pronounced gains in sparse-object settings, as indicated by our cross-module ablations. We will accordingly revise the relevant descriptions in the Abstract, Introduction, Method, Experiments, and Appendix accordingly, so that the manuscript consistently presents the complementary yet asymmetric roles of LAGM and GISA. We will also refine the contribution framing to state this point explicitly: LAGM and GISA play complementary but asymmetric roles, rather than both addressing serialization to the same extent.
> > >
> > > We sincerely thank the reviewer again for this careful clarification. We will revise the final version accordingly based on the reviewer's suggestions, and we believe these suggestions will help us present the paper in a more precise and better calibrated way.

---

### Official Review · Reviewer_JJKd · 2026-03-12

**Soundness:** 2
**Presentation:** 4
**Significance:** 3
**Originality:** 3
**Overall Recommendation:** 4
**Confidence:** 3

**Summary:**

This work, identifies an underlying problem of current State-Space Models that process 3D point clouds, namely their requirement to impose a 1D sequence structure in the inherently unordered 3D point clouds. The authors argue that such a requirement creates a serialization bias that can hurt the performance and generalization of the trained networks. To address that, they propose to split the processing into two distinct steps: first a local geometric encoder (LAGM) that learns anisotropic geometric features using ellipsoidal spectral encodings, and second, an Isotropic Semantic Aggregation (GISA) that uses a  sequential aggregation mechanism that can support both an order-independent (Identity Mode) and order-dependent (Morton Mode)  accumulation. The authors discuss how these modes depend on the sparsity of the different types of input point clouds used to train and test the model. The authors perform an extensive evaluation and ablation studies of the proposed method, showing both how the individual contributions affect the performance of the model and how the overall architecture is able to outperform previous state-of-the art methodologies.

**Compliance With Llm Reviewing Policy:**

Affirmed.

**Final Justification:**

The rebuttal of the authors addressed my concerns regarding their claims on exact invariance. The authors provided clarifications and more precise statements that make the contribution of their proposed method clear. Thus I am raising my recommendation to weak accept.

**Key Questions For Authors:**

- Is there any guarantee or experimental evidence that $\alpha_t$ goes to 0 in the case of the identity mode? Also, what is the rate of this convergence to 0?
- Given that the output $o_t$ only depends on the previous aggregated states how this results in ordering independence even when $\alpha_t=0$
-  In the identity mode, what is the mechanism by which the data loading ordering is chosen?

**Limitations:**

yes

**Strengths And Weaknesses:**

Strengths:
- This work provides a valuable insight into a known underlying problem of SSM methods for 3D point clouds, namely the introduced sequential bias, and proposes a novel alternative that aims to address it.
- The supporting experimental evaluation demonstrates how allowing the model to remove the sequential dependencies in the processing of the input data, allows it to outperform current state-of-the-art methods. This strengthens the claims of the authors about both the significance of the unidentified problem in the previous method and also the ability of their proposed model to partially address such a problem.

Weaknesses:
- While the authors claim that the proposed method allows for order-independent aggregation when $\alpha_t$ goes to 0, they don’t provide any study showcasing the actual behavior of $\alpha_t$ in the point clouds where the best identified mode is the identity mode. As a result, it is not clear if the improved performance originates from the order-independent aggregation that the authors claim to achieve or from some other underlying reason.
- Additionally, even when $\alpha_t=0$ and the aggregation is order independent, the output at each point still depends on the aggregated state of only the previous points in the sequence, making it dependent on the specific ordering in which the parallel summation happened. This, along with the previous weakness, doesn’t support the overall claim of the authors regarding the removal of the sequential bias
-  In the Identity mode, the authors claim the ordering of the points is according to the data loading ordering without discussing the specific mechanism by which this data loading ordering is computed. Is this the order in which the points were recorded by a scene scan? If so, different scanning strategies may introduce a sequence bias that is not apparent to the user, and it is not discussed in the paper

---

> ### Author Rebuttal · Authors · 2026-03-31
>
> We sincerely thank the reviewer for the thoughtful and encouraging reviews. We have addressed all comments, provided extensive new experimental validations, and clarified theoretical details below.
>
> **1. Is there evidence that $\alpha_t$ approaches zero in Identity Mode, and what is the convergence rate?**
>
> We agree our original wording "$\alpha_t \to 0$" was not precise enough. Because $\alpha_t$ is an input-dependent, dimension-wise parameter learned per position, it is not a globally controlled scalar and therefore does not have a single prescribed convergence rate. Our intended claim is narrower: relative to Morton Mode, Identity Mode in dense scenes exhibits significantly weaker decay and higher retention.
>
> To demonstrate this, we define the per-step retention coefficient as $r_t = \exp(-\alpha_t)$, which is the coefficient of the previous state $s_{t-1}$ in Eq. (8). Thus, smaller $\alpha_t$ means larger $r_t$ and stronger retention. We summarize the distribution of $r_t$ in Table 1, where $r_t.mean$ denotes average of $r_t$, and `frac` represents the proportion of $r_t$ in different intervals.
>
> Table 1 does not support a literal uniform convergence to zero, but it clearly demonstrates a dataset-dependent retention regime. In dense scenes (S3DIS / identity), the model actively shifts mass toward high retention, with 41.84% of activations maintaining $r_t > 0.9$ ($\alpha_t \to 0$) (effectively weak or no decay). When forced to use Morton mode, this high-retention mass drops drastically to 13.23%. Conversely, for sparse objects (ScanObjectNN), all modes concentrate in the low-retention regime ($r_t \le 0.1$) to preserve necessary spatial locality. We will revise the text to replace the absolute "$\alpha_t \to 0$" with the more precise statement of "weaker decay and higher retention in dense scenes."
>
> Table 1. GISA retention statistics.
>
> | Setting | $r_t.mean$ | `frac`（$r_t$<=0.1）| `frac`（0.1<$r_t$<=0.9） | `frac`（$r_t$>0.9）|
> |---|---:|---:|---:|---:|
> | `S3DIS / identity` | 0.4313 | 55.18% | 2.97% | 41.84% |
> | `S3DIS / morton` | 0.1358 | 86.00% | 0.77% | 13.23% |
> | `ScanNetV2 / identity` | 0.3006 | 67.91% | 3.73% | 28.36% |
> | `ScanNetV2 / morton` | 0.1677 | 82.39% | 1.72% | 15.89% |
> | `ScanObjectNN / identity` | 6.03e-06 | 100.0% | 0.00% | 0.00% |
> | `ScanObjectNN / morton` | 5.94e-05 | 99.998% | 0.002% | 0.00% |
>
> **2. If $o_t$ depends on previous aggregated states, why can the method still be described as order-independent when $\alpha_t = 0$?**
>
> We agree  strict prefix-wise invariance does not follow from this derivation, even when $\alpha_t = 0$. In that case,$s_t = s_{t-1} + \beta_t \odot (k_t \odot v_t),$so the final state $s_N = \sum_i \beta_i \odot (k_i \odot v_i)$ is commutative,
> but the intermediate output$o_t = q_t \odot s_t = q_t \odot \sum_{i \le t} \beta_i \odot (k_i \odot v_i)$still depends on the prefix state. Accordingly, our claim should not be read as strict invariance of every intermediate output. For unordered point clouds, the more relevant notion is robustness of final point-wise predictions after aligning outputs to the same point identities.
>
> Table 2. Direct order-perturbation test on ScanNetV2.
>
> | Input order | mIoU |
> |---|---:|
> | original | 78.47 |
> | reverse | 78.43 |
> | lex | 78.69 |
> | morton | 78.39 |
> | hilbert | 78.65 |
> | random × 5 | 78.49 ± 0.12 |
> | chunk_shuffle × 5 | 78.46 ± 0.10 |
>
> Here, we keep model weights fixed and perturb only the inference-time input order. `original` denotes the default preprocessing/loading order; `reverse` reverses it; `lex` sorts by `(x,y,z)`; `morton/hilbert` apply those serializations only at inference; and `random × 5` / `chunk_shuffle × 5` report mean ± std over five runs. The results remain highly stable across substantially different input orders: the best-worst gap is only `0.30` mIoU, and the random/chunk-shuffle fluctuations are also small. This does not establish strict invariance of every intermediate state, but it supports a narrower empirical claim: Identity Mode avoids explicit geometry-driven serialization priors, and in dense scenes the final predictions remain robust to input order.
>
> **3. In Identity Mode, how is the data-loading order chosen?**
>
> Identity Mode does not mean that order disappears in a mathematical sense; rather, it means that we do not apply geometry-based re-serialization. In our implementation, GISA receives the default point order after preprocessing, regrouping, downsampling, and data loading, rather than an explicit Hilbert/Morton traversal. In validation/test, the loader uses `shuffle=False` together with deterministic preprocessing and downsampling. Accordingly, Identity Mode is more accurately described as `default preprocessing/loading order without geometry-based re-serialization`.This order is neither the scanner’s raw acquisition order nor a theoretically unique canonical order; the precise claim is simply that Identity Mode avoids injecting a traversal prior constructed from geometric coordinates.

---

> > ### Author Rebuttal · Reviewer_JJKd · 2026-04-04
> >
> > I thank the authors for their thorough response to my questions. Most of my concerns are resolved and I am happy to increase my recommendation

---

> > > ### Author Response · Authors · 2026-04-04
> > >
> > > We sincerely thank the reviewer for the thoughtful follow-up comments. We are grateful that our additional clarifications and experiments have adequately addressed the concerns, and we greatly appreciate the reviewer's encouraging reassessment.

---

### Official Review · Reviewer_WmZq · 2026-03-20

**Soundness:** 2
**Presentation:** 2
**Significance:** 2
**Originality:** 2
**Overall Recommendation:** 4
**Confidence:** 3

**Summary:**

This paper proposes a linear serialization in 3D point clouds for 3D understanding tasks. The authors argue that it is necessary to design different embedding designs in local geometry and global semantic space. In the local geometry, point locations and surface normals are highly diverse and different, however in the global context, the 'chair' itself is a chair regardless of its locations.

Based on this motivation, the paper proposes
- Sec3.2 LAGM: Local Anisotropy Geometric Modeling
- Sec3.3 GISA: Global Isotropy Semantic Aggregation

Overall, the performances of this study is highly promising in series of benchmarks: ScanObjectNN, S3DIS Area 5, and ScanNetV2.

**Compliance With Llm Reviewing Policy:**

Affirmed.

**Final Justification:**

The authors well resolve my concerns. I do agree with the authors' claims. So my final score is `weak accept`.

**Key Questions For Authors:**

### Q-1. About weaknesses

I want to read the rebuttals about W-1.

### Q-2. Clear ablation study.

Also, I want to see the performances without using the two modules, LAGM and GISA, across different benchmarks. I feel like several hyper-parameters and design choices dominantly affects the performance of the proposed model. So, it will be great to see the results.

### Q-3. Theoretical analysis

Regarding Ellipsoidal Spectral Encoding, I wonder what is the difference between positional embedding in NeRF and the proposed Ellipsoidal Spectral Encoding? In my understanding, the only difference is whether to encode absolute locations (NeRF) or relative locations (the proposed method). Moreover, it is also highly similar to the Rotary Position Encoding (RoPE). It should be better to provide comparisons, and theoretical analysis when it applies to the scaled dot product attention. For example,

(F_q + R_q) \cdot (F_k + R_k)^T = ...
where R_q \cdot R_k = \psi(q) \cdot \psi(k)  // following the eq. 1 in the main paper.

Please refer to the equations 7 and 8 in the recent paper, `Rotary Position Embedding for Vision Transformer` which is cited in DINOv3 and SAMv2, etc.

By the way, if it is not that meaningful or impossible, please let me know.

**Limitations:**

Overall, I cannot catch the importance of the addressed problem by this paper. Existing methods from image/video domain do not really care about 'local geometry' and 'global semantics' in terms of embedding designs. I wonder why it is highly needed in 3D vision. Simply applying linear serialization is still okay for me though it is determinstic.

Also, if possible, I want to know the theoretical analysis of the embedding designs in two modules.

**Strengths And Weaknesses:**

### S-1. The motivation is clear.

I do understand that the local geometry and the global semantics are different. So the authors want to deal with this by newly designing the `position embedding modules` (<-- not sure this is a correct term to cover LAGM and GISA).

### S-2. Experiments results are admirable.

The authors evaluate the proposed modules on three different benchmarks including object-level points and scene-level points. This is the common setup, so it is okay. Moreover, the method achieves SoTA performances. I do not follow up the most recently released papers. But, as far as I know, the paper covers lots of popular baselines. So, for me, it is okay.

### W-1. Is it novel to address this issue?

I do not think that the embedding design problem from low-level vision signal to high-level semantic spaces is limited to the 3D vision. When we thinking of the series of recent studies in the image/video domains, I do not think that the addressed problem itself is novel. For example in DINOv3, the architecture itself is highly simple. They used Vision Transformer, with simple positional embeddings. Moreover, the 2D Rotary Position Encoding (2D RoPE) is applied at every scaled-dot product attention operator within each transformer layer. Even though the method really cares about 'local pixel signal' and 'global semantics', the method itself properly learns to predict semantic embeddings at the end. From this observation, the addressed issue by the authors -- discrepancy between local embeddings and global semantics -- are not that significant problems in the architecture design.

While the authors mention that pointclouds are not easy to proceed with the linear serialization, I believe that the video domain also have the same problems. While there are overlapping patches in different frames, as far as my understanding, lots of Mamba-based video architectures do not really typically address this issue.

Theoretically speaking, I believe that the authors' claims are right. However, I do not think that aforementioned claims are not that highly novel to me.

---

> ### Author Rebuttal · Authors · 2026-03-30
>
> We thank the reviewer for the helpful comments. Q1-Q3 ask whether our framing is necessary, empirically supported, and theoretically well positioned. We clarify below.
>
> **1. Why is this issue particularly important in 3D linear/SSM modeling?**
> We agree that this issue is not unique to 3D. Our claim is narrower: it becomes especially severe in 3D point clouds because they are both large-scale and unordered. Unlike images, point clouds have no regular grid; unlike videos, they have no natural temporal axis. Therefore, there is no canonical traversal order aligned with the data itself.
>
> To handle this scale and irregularity, recent 3D methods such as PTv3 introduce serialization strategies to improve efficiency, but such approaches still do not avoid the quadratic complexity of attention. This is why the community has also begun to explore linear-complexity SSM/Mamba-style models in 3D. The difficulty is that these models are fundamentally 1D sequence aggregators: before their update rules can operate, the unordered 3D point set must first be mapped into a 1D traversal, typically through Morton/Hilbert or related curves, as in *PointMamba* and *Point Cloud Mamba*. As a result, nearby 3D points may become distant in the sequence, while distant points may become sequentially adjacent.
>
> More importantly, the same imposed traversal can affect both local geometric modeling and global semantic aggregation. It influences which points are treated as local neighbors and, because the state update is path-dependent, also biases how global information propagates. Figure 3 gives a concrete example: with the same query point, `Identity Mode` yields more concentrated responses on semantically consistent regions, whereas `Morton Mode` shows stripe-like high-response bands on bookshelf and nearby planar regions. These patterns reflect serialization-induced feature coupling rather than true 3D neighborhood structure.
>
> Therefore, our contribution is not to claim that local-to-global modeling is uniquely a 3D problem, but that, in unordered 3D point clouds with linear/SSM aggregation, local anisotropic geometry and global semantic accumulation should not be forced to share the same serialization assumption. This is why we decouple them in LAGM and GISA.
>
> **2. Are the ablations sufficiently clear across benchmarks?**
> We agree that the original ablation in Table 4 was not sufficiently clear across benchmarks. We therefore report the supplementary results below in the same module-wise form for ScanNetV2 and ScanObjectNN.
>
> Table 1. Cross-module ablation on ScanNetV2.
>
> | LAGM | GISA | Mode | mIoU |
> |---|---|---|---:|
> | ✗ | ✗ | MLP | 75.88 |
> | ✓ | ✗ | MLP | 77.25 |
> | ✗ | ✓ | `Identity` | 76.87 |
> | ✓ | ✓ | `Identity` | 78.52 |
>
> Table 2. Cross-module ablation on ScanObjectNN.
>
> | LAGM | GISA | Mode | OA | mAcc |
> |---|---|---|---:|---:|
> | ✗ | ✗ | MLP | 90.35 | 89.08 |
> | ✓ | ✗ | MLP | 91.53 | 90.64 |
> | ✗ | ✓ | `Morton` | 93.06 | 92.81 |
> | ✓ | ✓ | `Morton` | 94.21 | 93.41 |
>
> These results suggest that the gains are not dominated by incidental hyper-parameters or auxiliary design choices: removing either module causes a drop, and the best-performing setting in each benchmark uses both modules.
>
> **3. How should ESE be positioned relative to NeRF PE and RoPE?**
> NeRF PE, RoPE, and ESE differ not only in using absolute versus relative coordinates, but in what they encode.
>
> NeRF-style positional encoding represents absolute spatial position by mapping a point coordinate $(x,y,z)$ into a higher-frequency space; it mainly answers where a point is. RoPE represents relative position inside attention through the query-key interaction, so its effect is naturally analyzed through the $QK^\top$ score; in this sense, it answers who is where relative to whom. ESE instead encodes local geometry through relative offsets within k-NN neighborhoods, i.e., what structure the neighborhood forms around a point.
>
> For ESE versus NeRF PE, the key difference is therefore not simply absolute versus relative coordinates, but generic positional representation versus anisotropic local geometry modeling. Concretely, ESE does not treat the local offset $\Delta p_{ij}$ isotropically; it assigns different sensitivities to different spatial directions, yielding an ellipsoidal rather than isotropic local receptive field. This is also consistent with our ablations, where replacing the isotropic local variant with ESE yields consistent gains across benchmarks.
>
> For ESE versus RoPE, the distinction is also direct. As discussed in *Rotary Position Embedding for Vision Transformer* (Eqs. (7)(8), Sec. 3.2), RoPE acts through the query-key interaction inside attention, so the reviewer’s suggested $QK^\top$ analysis is natural for RoPE. Our ESE does not operate in this way: it acts before neighborhood aggregation by changing how local neighbors are represented. Therefore, the $QK^\top$ framework is natural for RoPE, but not the most direct analysis for ESE.

---

> > ### Author Rebuttal · Reviewer_WmZq · 2026-04-06
> >
> > Overall, the authors resolved my concerns.
> > - I do understand the difference between the proposed PE against NeRF's PE, RoPE, etc.
> > - The authors provide the cross-module ablation study
> > - The authors summarize the necessity of the linear attention (SSM style).
> >
> > Also, I confirmed that the other reviewers gave the plausible score. Accordingly, I will also increase my score the borderline. I believe that it is okay for me to give acceptance for this submission.

---

> > > ### Author Response · Authors · 2026-04-07
> > >
> > > We sincerely thank the reviewer for the very encouraging follow-up. We are grateful that the clarifications on ESE vs. PE/RoPE, the added cross-module ablations, and the discussion on the necessity of linear attention have fully addressed the concerns, and we truly appreciate the reviewer's positive reassessment of our submission.

---

### Decision · Program_Chairs · 2026-04-30

**Decision:**

Accept (regular)

**Comment:**

This submission eventually got four positive recommendations. Initially, the reviewers were concerned about the novelty, the technical design, and the performance. The authors did a good job and addressed most of these concerns in the rebuttal. During the discussion among the authors and the reviewers, the reviewers confirmed that their concerns had been fully addressed. Thus, all reviewers reached a consensus without a discussion. The AC read through the manuscript, all reviews, the rebuttal, the discussions among the authors and the reviewers, and the author AC confidential comment, the AC agreed with all reviewers, and liked the idea of the paper. Per these, the AC made a decision of acceptance. This decision was approved by the SAC as well.